# Imaging in Transcatheter Mitral Valve Replacement: State-of-Art Review

**DOI:** 10.3390/jcm10245973

**Published:** 2021-12-20

**Authors:** Manuel Barreiro-Perez, Berenice Caneiro-Queija, Luis Puga, Rocío Gonzalez-Ferreiro, Robert Alarcon, Jose Antonio Parada, Andrés Iñiguez-Romo, Rodrigo Estevez-Loureiro

**Affiliations:** Cardiology Department, University Hospital Alvaro Cunqueiro, Galicia Sur Health Research Institute (IISGS), 36213 Vigo, Pontevedra, Spain; bcanque@gmail.com (B.C.-Q.); luis.romeu.puga@gmail.com (L.P.); ferreiro_44@hotmail.com (R.G.-F.); dr.alarcon1587@hotmail.com (R.A.); chechocat94@gmail.com (J.A.P.); andres.iniguez.romo@sergas.es (A.I.-R.); roiestevez@hotmail.com (R.E.-L.)

**Keywords:** structural heart intervention, transcatheter mitral valve replacement, mitral regurgitation, transoesophageal echocardiography, cardiac computed tomography

## Abstract

Mitral regurgitation is the second-most frequent valvular heart disease in Europe and it is associated with high morbidity and mortality. Recognition of MR should encourage the assessment of its etiology, severity, and mechanism in order to determine the best therapeutic approach. Mitral valve surgery constitutes the first-line therapy; however, transcatheter procedures have emerged as an alternative option to treat inoperable and high-risk surgical patients. In patients with suitable anatomy, the transcatheter edge-to-edge mitral leaflet repair is the most frequently applied procedure. In non-reparable patients, transcatheter mitral valve replacement (TMVR) has appeared as a promising intervention. Thus, currently TMVR represents a new treatment option for inoperable or high-risk patients with degenerated or failed bioprosthetic valves (valve-in-valve); failed repairs, (valve-in-ring); inoperable or high-risk patients with native mitral valve anatomy, or those with severe annular calcifications, or valve-in-mitral annular calcification. The patient selection requires multimodality imaging pre-procedural planning to select the best approach and device, study the anatomical landing zone and assess the risk of left ventricular outflow tract obstruction. In the present review, we aimed to highlight the main considerations for TMVR planning from an imaging perspective; before, during, and after TMVR.

## 1. Introduction

Mitral regurgitation (MR) is the second-most frequent valvular heart disease encountered in clinical practice in Europe [1], and it is associated with high morbidity and mortality [2]. Recognition of MR should encourage the assessment of its etiology, severity, and mechanism in order to determine the best therapeutic approach [3].

Mitral valve surgery constitutes the first-line therapy for patients with symptomatic severe MR [3]; however, up to 50% of those affected are not referred for surgery due to high risks [4].

In recent years, transcatheter procedures have emerged as an alternative option to treat inoperable and high-risk surgical patients [5]. The edge-to-edge leaflet repair system (TEER) represents the most frequently applied percutaneous transcatheter mitral valve procedure. In patients with suitable anatomy, it can be successful and safe [6]. The current European Valvular Heart Disease Management guidelines [3] give Class IIb recommendations for transcatheter mitral valve repair in symptomatic patients with severe primary MR despite optimal medical therapy, reasonable life expectancy but prohibitive surgical risk; and Class IIa recommendations for symptomatic patients with severe secondary MR fulfilling the anatomical inclusion criteria who are not eligible for surgery. However, due to the complexity and heterogeneity of mitral valve anatomy and pathology, some patients do not meet the eligibility criteria for TEER and repair may be ineffective (rheumatic etiology, endocarditis-related valve disease, prior MV surgery, cleft or perforated mitral leaflets, lack of secondary chordal support, posterior leaflet length < 7 mm, leaflet gap > 2 mm, presence of severe calcifications in the grasping area, transmitral pressure gradient > 4 mmHg or MV area < 3.5 cm^2^) [7].

Transcatheter mitral valve replacement (TMVR) has appeared as a promising intervention that may overcome some of the current limitations associated with TEER [8]. However, some limitations, such as apical access and the associated thoracotomy marked early experiences with TMVR. The development of transseptal TMVR, by means of improved technology in delivery systems, has allowed TMVR to grow. Transseptal access has shown that it is effective, safe, and also offers less morbidity and recovery time compared to the trans-apical approach [9]. Thus, currently TMVR represents a new treatment option for inoperable or high-risk patients with degenerated or failed bioprosthetic valves, valve-in-valve (ViV); failed repairs, valve-in-ring (ViR); inoperable or high-risk patients with native MV anatomy, or those with severe annular calcifications, or valve-in-mitral annular calcification (ViMAC) [10].

Despite the advancements, TMVR implies a not negligible risk of periprocedural and post-procedural complications [11], and still faces significant disadvantages [12]. The procedure is still not suitable for all, and the most common causes of TMVR exclusion are frailty, severe tricuspid regurgitation, prior aortic valve therapy, mitral anatomical exclusion, severe MAC, and the risk of left ventricular outflow tract (LVOT) obstruction [12].

In the present review, we aimed to highlight the main considerations for TMVR planning from an imaging perspective. This study reviews the role that multimodality cardiac imaging plays before, during, and after TMVR.

## 2. Imaging Overview

Advances in imaging have enabled the TMVR technique to evolve. Cardiovascular imaging has become a key player in diagnosis, pre-procedural planning, procedural guidance, and follow-up in TMVR therapies. Moreover, a patient-centered structural intervention team with the interventional and the imaging parties well familiarized with each other’s tools, skills, language, and procedures are essential for a successful intervention [10].

A pre-procedure cardiac imaging examination, through multimodality imaging, is crucial to identify the severity, etiology, and mechanisms of MR; the coexistence with any degree of mitral stenosis or any other valvular abnormality, and to determine patient eligibility according to the anatomic measurements and anatomic variables used for every specific device. Also, the pre-TVMR cardiac imaging examination should help to predict the risk of potential procedural complications and their likelihood and to localize the most suitable points for access and puncture [12].

Pre-procedural transthoracic echocardiography (TTE) is mandatory and should be the first cardiac imaging examination for patients with a suspicion of mitral valve disease, as it is noninvasive and provides a first characterization of the magnitude and etiology of the mitral valve disease.

Beyond TTE, both transesophageal echocardiography (TEE) and cardiac computed tomography (CCT) modalities are the cornerstones for successful TMVR procedures [13,14]. TEE has the superiority of temporal resolution, hence, is the method of choice for mitral valve function and leaflet characterization. On the other hand, CCT is a non-invasive imaging technique with high isotropic spatial resolution and excellent calcification definition, offering ideal capabilities for a higher accuracy for 3D sizing and procedural simulation [10]. This multimodality imaging approach is, at the time, the gold standard for TMVR [15]. Table 1 shows the advantages and the preferred method for screening, peri-intervention assessment, and post-procedural follow-up.

Echocardiography screening is the first step to assess the indications for a valvular intervention. It includes characterization of the valvular disease mechanism, grading, as well as its impact on heart size and function. Moreover, evaluation of right heart cavities and pulmonary hypertension are important prognostic factors that should be noted [10]. Potential contraindications should also be sought, such as active endocarditis, intracardiac thrombus, or severe patient-prosthesis mismatch [16]. Determining the acoustic window quality and optimizing patient position are also important steps since procedural guidance relies on TEE imaging. 3D-TEE with multiplane reconstruction is a vitally important tool for the correct assessment of valvular or prosthetic valve anatomy, although acoustic shadowing due to extensive calcification, prosthetic heart valves, or annuloplasty rings may hinder a complete analysis of sub-valvular apparatus or LVOT. During the procedure, the echocardiographer will provide continuous image guidance with TEE in close collaboration with the interventional team. Bicaval, aortic short-axis and four-chamber views may help to select the appropriate septal puncture site (the ideal position usually slightly superior and posterior from the midpoint of the interatrial septum). TEE is also used to guide the advancement and positioning of the TMVR prosthesis within the native MV annulus. Simultaneous bicommissural-LVOT and 3D views are highly valuable for final adjustments, which are performed based on TEE image. Immediately after TMVR deployment TEE may help to assess perivalvular leak (PVL), residual MR, mitral gradients, rule out LVOT obstruction and gradients measurements.

Cardiac computed tomography (CCT) is considered to be essential for TMVR planning. Contrast-enhanced thin-sliced electrocardiography-gated CCT is mandatory. The use of retrospective gating covering the whole cardiac cycle with a 5–10% R-R interval reconstruction is highly recommended, and mandatory to cover the whole systolic phase [17]. CCT offers an isotropic sub-millimeter spatial resolution, facilitating accurate mitral geometry assessment and annular sizing [12]. CCT is employed to evaluate patient suitability according to all TMVR systems’ official recommendations. There are some common anatomic points routinely evaluated for all TMVR valve systems, although other CCT-based measures are device-specific, leading to different CCT workup and evaluation algorithms for each valve system. The most relevant aspects of CCT evaluation before TVMR are mitral annulus measurements (intercommissural and anterior-posterior diameters, inter-trigone distance, perimeter, area and calcification assessment), mitral leaflets (length, thickness and calcification), interatrial septum anatomy, left atrial and left ventricle anatomy and LVOT characteristics (aorto-mitral angle, baseline area at systole and diastole and neo-LVOT assessment after virtual valve implantation) [10,12].

CCT also provides a detailed and clear definition of the extent and severity of annular calcium. Some measures such as maximal height and thickness of the observed calcification, the circumferential extension and trigone and leaflets involvement are used for the planning and stratification of TMVR embolization risk.

LVOT obstruction following TMVR is one of the most feared, and potentially fatal, complications. Therefore, recommendations have been issued regarding neo-LVOT estimation to screen and prevent this complication [18]. The neo-LVOT is the result of the dislodgment of the anterior leaflet of the mitral valve toward the ventricular septum [19]. The CCT virtual valve implantation and the evaluation of the neo-LVOT area on a 3D dedicated software best predicts the risk of LVOT obstruction (Figure 1). The predicted neo-LVOT is measured at mid-late-systole as the narrowest 2-dimensional area between the virtual valve and the ventricular septum [20]. Predicted neo-LVOT area < 200 mm^2^ identifies patients at risk of significant LVOT obstruction; and a neo-LVOT area < 170 mm^2^ has been shown to predict LVOT obstruction with 96.2% sensitivity and 92.3% specificity. Other observed features related to LVOT obstruction are the presence of a bulky septum (>15 mm thickness or <17.8 mm annulus-to-septal distance), an acute aorto-mitral angle (<110°), an elongated anterior mitral leaflet (>25 mm) and the presence of left ventricle small cavity size (end-diastolic diameter <48 mm), hypertrophy (LV mass index >105 g/m^2^) or preserved ejection fraction [14,18]. Preemptive LVOT obstruction avoidance strategies have been reported in selected high-risk cases such as alcoholic septal ablation or LAMPOON techniques (base-to-tip [21]; tip-to-base or reverse LAMPOON [22], or anterograde LAMPOON [23]), although data regarding outcomes in large series are missing. A pre-procedural LVOT management algorithm has been recently published [19].

Furthermore, it facilitates procedure planning allowing for fluoroscopic projection estimation (en-face, two-chamber and three-chamber views) and access planning (Figure 2). CCT may help to select the most suitable location for transeptal (distance to mitral annular plane, thickness and morphology) or transapical puncture site (most appropriate intercostal space, distance from apex to mitral annular plane and trajectory avoiding any disturbance with papillary muscles). An abdominal-pelvic venous phase CT scan may be useful to evaluate vein diameters and tortuosity for a transeptal approach case.

## 3. TMVR: ViV, ViR, ViMAC

Reoperation in degenerated mitral surgical heart valves (SHV) or in failed surgical repair has a high mortality and morbidity risk. TMVR has demonstrated good outcomes for degenerated bioprosthetic valves (ViV) and acceptable results in failed mitral repair (ViR); making adequate patient selection, pre-procedural planning, and operator experience necessary. Transcatheter valve-in-valve implantation in the mitral and tricuspid position may be considered in selected patients at high risk for surgical reintervention according to the actual European guidelines [3].

Mitral annular calcification (MAC) is a degenerative age-dependent process leading to MR or mitral stenosis in severe cases. It has been linked to cardiovascular risk factors and other pathologies [24]. MAC patients tend to be poor candidates for mitral surgery due to technical challenges and the risk of complications.

Currently, experiences have been described with MAC disease using aortic THV and dedicated mitral THV devices [25].

### 3.1. Procedural Description

Procedural steps are described in detail in the literature [26]. Briefly, the TMVR procedure is usually performed under general anesthesia with TEE and fluoroscopic guidance. Regarding approaches, the transseptal and transapical represented the preferred ones. For vascular access, there is a general consensus that ultrasound guidance is considered the standard of care [27].

There is a growing interest in the transseptal approach, as it is the less invasive option. The anatomic target for the transseptal puncture varies by procedure [28]. In general, the preferred transseptal site puncture for TMVR procedures is mid-to-superior and posterior to the center of the fossa ovalis (approximately 3.5–4.0 cm over the mitral plane). Once the sheath enters the left atrium a 0.032-inch exchange wire is placed in the upper left pulmonary vein, if possible. Next, crossing the mitral valve is facilitated by the flexible Agilis catheter (St Jude Medical, St Paul, Minnesota) using a 5-Fr diagnostic catheter mounted on a standard 0.035-inch exchange wire. Then, a pigtail catheter is delivered into the left ventricle and a J-preshaped stiff wire (such the Safari wire, Boston Scientific) is advanced through. Afterward, the Agilis catheter is withdrawn and the atrial septum is dilated using 12–16 mm peripheral balloons. For the transseptal approach, the SAPIEN (Edwards Lifesciences, Irvine, CA, USA) valves are the most used transcatheter heart valves (THV). The SAPIEN 3, with a lower profile and smaller sheath, provides several advantages. In this case, the THV prosthesis must be mounted for antegrade implantation. Septal crossing is usually done under fluoroscopic and TEE guidance with no push. Positioning THV is executed in the projection perpendicular to the plane of the mitral annulus, carefully advancing the valve near the mitral orifice with the objective of 20–30% of the THV toward the left atrium and 70–80% toward the left ventricle. The implantation depth is adjusted so the external skirt of SAPIEN 3 connects throughout the landing zone. A more ventricular final position may provide better hemodynamic performance with less valvular gradient, but a higher risk of LVOT obstruction. On the other hand, a more atrial final position may provide lower neo-LVOT gradients, but a higher residual paravalvular and prosthetic embolization likelihood. TEE guidance plays an important role to define the appropriate landing zone. A THV valve is deployed by slowly balloon inflation under rapid ventricular pacing (140 beats/min is usually adequate). Post-deployment assessment with TEE is required to confirm optimal function (presence, severity and mechanisms of PVL, transmitral gradients and leaflets motion).

On the other hand, transapical approach provides easy and direct access to the mitral valve. The procedure requires general anesthesia and a transapical approach through the left mini-thoracotomy. The procedure is mainly executed under TEE guidance. Pre-dilatation of the mitral valve apparatus with balloon valvuloplasty catheter is done at the discretion of the local team. A 34Fr sheath is advanced over a soft 0.035 wire into the left atrium. The implant device is advanced into the sheath and then positioned at the level of mitral annulus. Pacing is not needed for deployment in some dedicated mitral THV devices but is still necessary for aortic THV employed for TMVR.

### 3.2. Clinical Results and Published Evidence

Observational data for ViV TMVR has demonstrated good outcomes for degenerated bioprosthetic valves with adequate patient selection, pre-procedural planning, and operator experience. Transcatheter valve-in-valve implantation in the mitral and tricuspid position may be considered in selected patients at high risk for surgical reintervention according to the actual European guidelines [3]. However, TMVR for ViR and ViMAC is associated with a higher risk of procedural complications and increased mortality following TMVR compared to ViV.

Recently, data from the TMVR multicenter registry was published by Yoon et al. [29] evaluating procedural success and outcomes in this patient population. 521 high-risk patients (STS 9%) were evaluated, with 322 ViV patients, 141 ViR patients, and 58 ViMAC patients. The majority of access was transapical; however, 39.5% were transseptal. Ninety percent used the balloon-expandable Sapien valve. Technical success was 89.1%, and a second valve implant was most frequently needed in ViR followed by ViMAC and ViV (12.1%, 5.2%, 2.5%, respectively). At 30 days, there was a higher residual significant MR in ViR (18.5%) and ViMAC (13.8%) compared to ViV (5.6%) procedures, probably due to a higher rate of PVL after TVMR. Patients with residual MR are known to have higher mortality. All-cause mortality was lower in ViR (9.9%; 30,6%) and ViV (6.2%; 14,0%) at both 30 days and 1 year respectively; compared with worse results with ViMAC (34.5%; 62.8%) [30].

TMVR ViMAC early experience with off-label use of aortic balloon-expandable THV is exposed in two retrospective registries [30,31] showing high 30 day and one year mortality (25–35% and 54–63%, respectively). The first prospective, multicenter clinical trial for ViMAC using balloon-expandable aortic THV has been recently published [32] showing lower mortality rates at 30 day and one year than previously reported (6.7% and 26%, in order). Considering the Tendyne valve in MAC patients, initial experience has indicated high procedure success and without procedure mortality. Nonetheless, the authors recognized a highly selected patient population [33]. Unlike ViV and ViR, which are more consolidated procedures and included in clinical care and guidelines recommendations; ViMAC is in an early phase of development and it should be reserved for selected patients in highly experienced centers.

No significant difference in mortality, stroke, valve embolization or need for conversion to surgery was observed in transseptal compared to transapical access. However, TMVR via transseptal access was associated with a lower rate of life-threatening or fatal bleeding.

### 3.3. Imaging Key Aspects

During deployment of a THV within a surgical ring, bioprosthetic valve, or MAC, the principal imaging concerns are device size selection, implantation depth, device coaxially respect to mitral annulus, and complete expansion within the constraining tissue (native or prosthetic).

(I) Valve-in-Valve: The essential parameters by CCT are SHV dimensions assessment (internal diameter, height, projection into left ventricle), SHV tissue-type (lower risk of LVOT obstruction with porcine SHVs) and prediction of neo-LVOT area. The internal diameter of the surgical heart valve determined by CT scan helps to choose the optimal THV size because the goal is to achieve a conical shape of the THV after implantation [34,35]. CT measurements are highly dependent on image quality, acquisition and reconstruction technique, prosthetic material opacity, and associated blooming, as well as measurement technique; but a precise sizing of the landing zone decreases valve embolization or migration. CT imaging is helpful to confirm surgical heart valve (SHV) size or to establish SHV size in patients with an unclear surgical history. Imaging-derived measurements maybe not be equivalent to the stent’s true internal diameter, thus it can change for thickening and calcification of degenerated leaflets [36]. A smartphone app has been developed, and is available for different platforms, to assist SHV size selection before TMVR ViV [37,38].

The TMVR ViV procedure is guided by 3D-TEE (Figure 3) (transeptal puncture, coaxially alignment) and fluoroscopy (depth deployment). Immediately after THV deployment TEE is crucial to rule out LVOT obstruction, residual paravalvular regurgitation and THV hemodynamic performance (transvalvular gradient, intra-prosthetic residual regurgitation).

(II) Valve-in-Ring: There are multiple types of surgical MV annuloplasty rings and not all are suitable for a TMVR ViR procedure. To conform an acceptable landing zone, the surgical ring must become complete and circular or nearly circular. CT imaging is helpful to assess ring shape and type, internal dimensions (diameters, area and perimeter), leaflets calcification, length of anterior leaflet and predicted neo-LVOT area. It is important to note that, according to THV size selection, the ring shape may change from oval to circular after TMVR, increasing its area. The intraprocedural TEE monitoring is employed to guide the THV approach (transeptal puncture and alignment) and to exclude complications as ring dehiscence or anterior leaflet displacement into LVOT after THV deployment.

(III) Valve-in-MAC: CCT is complementary to echocardiography and has been the imaging modality of choice to evaluate patients for TMVR ViMAC [6]. *The appropriate pre-procedural patient selection for Valve-in-MAC requires expertise, is time-consuming and it has to be on consideration several anatomical aspects. In the previous published series, only 33% of evaluated cases for ViMAC were finally acceptable for the TVMR procedure* [20]. First, CCT evaluation of mitral calcification comprises (i) description of quality: brittle, caseous or vastly dense calcium; (ii) distribution: circumferential or noncircumferential; and (iii) severity (based on semiquantitative approach): fleck-like (mild), coalescing (moderate) and bulky/protruding (severe). Furthermore, to grade the severity of MAC and predict valve embolization, a CT-based score has been proposed [33]. A score ≥ 7 points defines severe MAC. The presence of bilateral commissural calcification, as well as some anterior calcification, provides a better anchoring for ViMAC; a recommendation of 270° of contact is considered sufficient to achieve complete sealing. Multi-intensity thick-slab projections facilitate anatomy understanding to trace the area and perimeter measures. Determination of the landing zone (contact between the THV and the annular calcification) is often done at mid-to-late systole by tracing a 3D ellipsoid at the leaflet-annular insertion [19]. It also requires 3D image simulation of the device implantation. The extension and severity of calcification on the mitral annulus are used to determine the degree of THV oversizing. Some authors recommend a 10–25% degree of oversizing to prevent PVL and late migration of the valve [14]. CCT is also fundamental to estimate the risk of LVOT obstruction. Predicted neo-LVOT area < 200 mm^2^ identifies patients at risk of significant LVOT obstruction demanding an adjunctive procedure, such as LAMPOON or septal reduction with transcoronary alcohol to ensure a safe procedure. Neo-LVOT area <100 mm^2^ identifies very high-risk patients where ViMAC should be avoided. Besides the neo-LVOT area, other anatomical features have been recently related to neo-LVOT obstruction after TVMR ViMAC; systolic LVOT area, indexed neo-LVOT, expected LVOT area reduction, and virtual THV to septum distance [39]. The procedure is guided by TEE and is highly valuable for ruling out complications after THV deployment as anterior leaflet displacement into LVOT, assessing the risk of embolization, or detecting residual paravalvular regurgitation.

## 4. Valve in Native Mitral Valve Replacement

TMVR on native anatomy has several challenges because the mitral valve apparatus is a very complex dynamic system involving several structures, interacting with the left ventricle, the left atrium and the aortic valve [40,41]. The first is related to the size of the mitral annulus, usually dilated in chronic MR. Complete sealing and stable anchorage of the prosthesis to prevent embolization or displacement represent major concerns of TMVR and pose a challenge due to the large anatomical variability between organic and functional MR. Furthermore, due to the proximity to the aortic valve and the LVOT, TMVR poses an important risk for LVOT obstruction, and is associated with poor clinical outcomes. There is a wide range of TMVR devices at various stages of development. Table 2 shows some TMVR for native anatomy devices with reported clinical data.

The most employed TMVR for native anatomy is the Tendyne device (Abbott, Menlo Park, California). This device is fully repositionable, retrievable and designed to be implanted using a transapical approach. The Tendyne system consists of two self-expandable nitinol frames (inner and outer stent) and a valve formed by three porcine pericardial tissue leaflets sewn onto the circular inner stent. The inner valve is sutured to the outer stent that is coated in porcine pericardium with a polyethylene terephthalate (PET) fabric cuff that provides the sealing surface within the native annulus. The outer stent is designed with a D-shape to fit the mitral annulus and facilitate the orientation of the straight edge against the aortic-mitral continuity. This prosthesis is sutured to an ultra-high molecular weight polyethylene tether designed to stabilize the valve after deployment, which is fixed to an epicardial pad of polyether ether ketone button covered in PET fabric through the left ventricular apex.

### 4.1. Procedural Description and Imaging Key Aspects

A standardized TEE and CCT evaluation of the mitral valve apparatus is required to determine anatomic suitability and appropriate valve sizing for Tendyne implantation with special attention to mitral annular dimensions (septal-lateral, inter-commissural dimensions and entire perimeter), left ventricular dimensions (measured in the 3-chamber view or the short axis view along the septal-lateral direction) and neo-LVOT evaluation (Figure 4).

The device is implanted under general anesthesia through a left mini-thoracotomy using a transapical approach, using 2D and 3D TEE imaging guidance. The access site and orthogonal annular trajectory are determined from pre-procedural CCT and intraoperative echo imaging. A standard 0.035-inch wire is inserted into the left atrium and a balloon tip catheter is advanced to the left atrium to ensure that the guidewire is not entrapped in the mitral subvalvular apparatus. A 34-Fr sheath is then placed over the wire into the left atrium. The valve prosthesis is delivered through the sheath and partially deployed in the left atrium, until the outer valve expands up to approximately 85% of its final size. The D-shaped outer stent is aligned with the straight edge oriented anteriorly against the aortic-mitral continuity by rotating the device, using TEE guidance. The delivery sheath is then retracted to deploy the remainder of the prosthesis in an intra-annular position. The length and tension of the tether are adjusted to optimize the seating of prosthesis for MR reduction and to minimize the risk of device displacement.

### 4.2. Clinical Results

The first two temporary valve implants were reported by Lutter et al. [42] in 2013 (before proceeding with conventional mitral valve surgery), and the first-in-human definitive implant was performed in 2014 [43]. Since then, the Tendyne system has accumulated the most extensive clinical data to date. The experience in the first 100 patients revealed promising results, with an implant success of 96%, with no need for emergency surgery or mortality during the procedure [44]. The 30-day mortality rate was 5.5% and the most frequent complication was hemorrhagic, at 20% of cases. At one year, mortality was 26% (cardiac death accounted for the majority of the deaths [22/26; 85%]), disabling stroke was 3%, and the need for reoperation to adjust the strap tension was 3%. There were no cases of embolization or device migration, although there was an incidence of 6% of device thrombosis (within the first 35 cases, when anticoagulation was not specified by the study protocol). MR was absent in 98.4% of patients at one year follow-up. No patients had LVOT obstruction or significant mitral stenosis. At one year improvement in symptoms and quality of life were evident: 88.5% of survivors were in NYHA functional class I or II (34.0% at baseline; *p* < 0.0001) and the KCCQ increased by ≥5 points in 81.3% and by ≥10 points in 73.4%. The device has also shown promise for the treatment of MR in the setting of severe MAC [36]. Nine patients were successfully treated, with relief of MR in all patients and without procedural deaths. At one year, the survival rate was 78% and the MR remained absent in all treated patients.

The SUMMIT trial (Clinical Trial to Evaluate the Safety and Effectiveness of Using the Tendyne Mitral Valve System for the Treatment of Symptomatic MR; NCT03433274) is investigating the safety and clinical benefits of the Tendyne system compared to the Mitraclip system in patients with symptomatic moderate-severe MR suitable for transcatheter edge-to-edge repair (randomized cohort). In addition, there are two other single-arm cohorts, which will evaluate the Tendyne system for the treatment of severe MR with or without MAC. Tendyne received CE mark approval in January 2020 (the first transcatheter mitral valve replacement device approved for clinical use in Europe).

Employing the Tendyne system, a new option has been reported for a failed TEER in a patient non-candidate for a new TEER procedure or MV surgery. The ELASTA-Clip is a feasible and safe transcatheter electrosurgical detachment of failed TEER clips from the anterior leaflet followed by Tendyne implantation [45].

Very recently, 30-day outcomes of an early feasibility trial with a novel TMVR system have been presented [46]. The Intrepid TMVR is a novel device designed in order to treat patients with severe MR through femoral access with 35Fr sheath. Initial results, despite including a very selected population, are promising. In a cohort of advanced-age patients with mainly primary MR and mildly impaired LVEF, there are no deaths, strokes or reinterventions at 30-days. Significant improvement in NYHA functional class has been also reported. Nonetheless, around 50% of patients had significant major bleeding events due to access site major vascular complications. There is a promising landscape for this device but at this moment only preliminary data are available.

## 5. Conclusions

TMVR represents a new treatment option for inoperable or high-risk patients with symptomatic severe MR in different anatomical scenarios (ViV, ViR, ViMAC, and native TMVR). Cardiac multimodality imaging (3D-TEE and CT) is crucial for detailed pre-procedural planning, intraprocedural monitoring and successful outcomes.

## Figures and Tables

**Figure 1 jcm-10-05973-f001:**
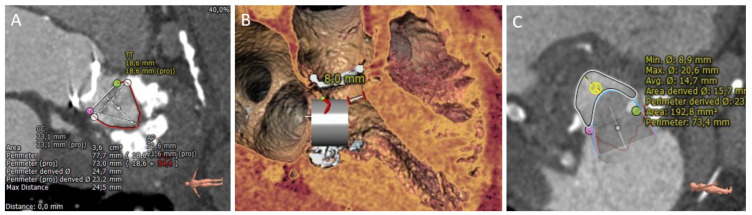
TMVR valve-in-MAC pre-procedural planning. (**A**) Mitral annular calcification with a 180° extension in the posterior and lateral aspect of mitral annulus. Internal dimensions can be noted on the image (TT: inter-trigone diameter; AP: anterior-posterior diameter; area and perimeter). (**B**) Three-dimensional virtual valve implantation (SAPIEN 3 23 mm) with a distance neo-valve to interventricular septum of 8 mm. (**C**) Neo-LVOT area according to the virtual valve implantation (Area 193 mm^2^).

**Figure 2 jcm-10-05973-f002:**
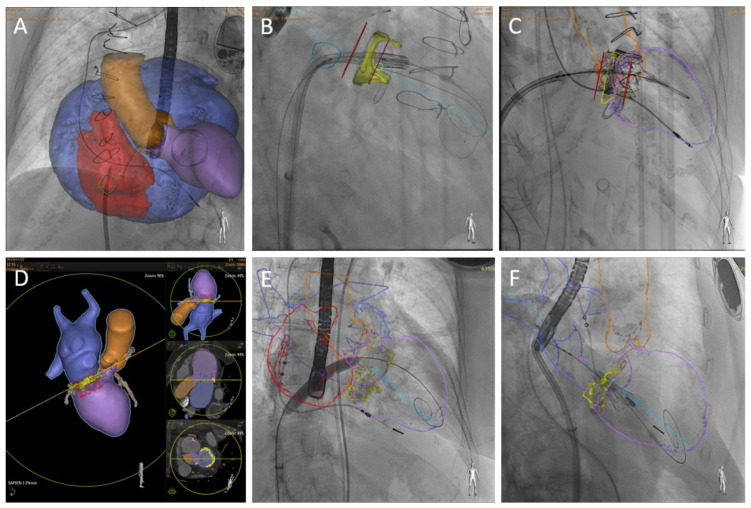
CT-fluoroscopy fusion imaging. The superior row shows a TMVR valve-in-valve procedure in a patient with extreme left atrium enlargement and modified projection required for transeptal puncture (**A**). Markers (red lines) may be over-imposed to fluoroscopy imaging to guide depth deployment (**B**,**C**). Inferior row, TMVR valve-in-MAC CT preprocedural planning (**D**), interatrial septal balloon dilatation (**E**) and initial phase of THV deployment with coaxial projection to mitral annulus (**F**).

**Figure 3 jcm-10-05973-f003:**
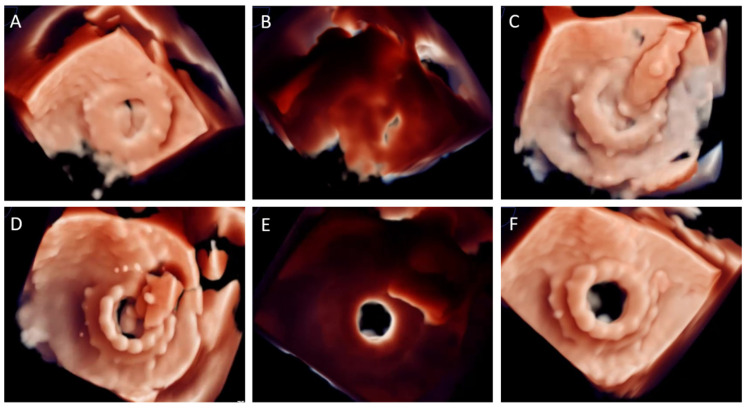
Three-dimensional transoesophageal echocardiography with photo-realistic rendering during TMVR valve-in-valve procedure. (**A**) En-face view of a degenerated mitral surgical prosthetic valve, with severe prosthetic stenosis. (**B**) Same image with light source place behind mitral prosthetic valve during diastole. Prosthetic leaflets thickening and mobility reduction can be easily noted. (**C**) THV positioning inside SHV. (**D**) Balloon-expandable THV deployment. (**E**) Immediate result after deployment. Same image configuration than (**B**), significant improvement in diastolic opening can be noted. (**F**) TMVR ViV final result en-face view.

**Figure 4 jcm-10-05973-f004:**
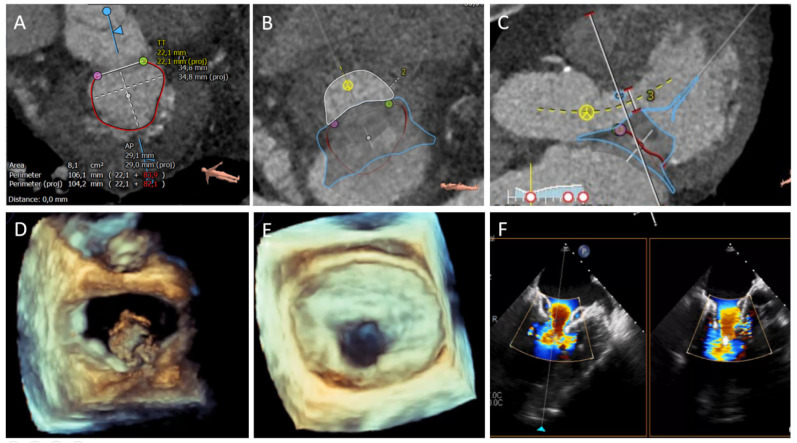
TMVR in native mitral anatomy with Tendyne (Abbott Medical) system. (**A**) Mitral annular dimension assessed with cardiac CT. (**B**,**C**) Neo-LVOT area after virtual valve implantation with specific Tendyne system design. (**D**) Three-dimensional TEE en-face view of initial THV device deployment and orientation. (**E**) Final result after complete deployment on 3D-TEE and in 2D-TEE color doppler on simultaneous bicommissural and LVOT views (**F**).

**Table 1 jcm-10-05973-t001:** Suggested assessment steps for TMVR with preferred modalities.

Assessment Steps	TEE	CT
Screening		
Valve disease mechanism	+++	+
Chambers size	+++	++
LV/RV function and pulmonary hypertension	+++	+
Valve disease grading	+++	-
Calcification extension	+	+++
Contra-indications assessment		
Endocarditis	+++	+
Thrombus	+++	+++
Severe patient-prosthesis mismatch	+++	-
Peri-intervention		
Vascular access	-	+++
Annulus sizing	++	+++
Fusion imaging	++	+++
Interatrial septum assessment/transeptal punction planning	+++	+++
Fluoroscopic projection estimation	-	+++
Neo-LVOT size estimation	+	+++
3D simulation/printing	+	+++
Procedural guidance/Device deployment	+++	-
Post-procedural		
Prosthetic valve function	+++	+
Paravalvular leak	+++	++
Vascular complications	-	+++

+++ Preferred method; ++ alternative method; + incomplete assessment; - not possible.

**Table 2 jcm-10-05973-t002:** Transcatheter mitral valve replacement devices.

Device	Intrepid	Tendyne	Tiara	EVOQUE	HighLife	SAPIEN M3
Patients, *n*	50	109	79	14	15	45
Etiology of MR						
Organic	16	11	8.9	28.6	27	55.6
Functional	72	89	62	21.4	73	35.6
Mixed	12		29.1	50		8.9
LVEF, %	43 ± 12	47.2	37 ± 9	54	38	44
Approach	TA	TA	TA	TF	TA	TF
Device implant success	98	97.2	92.4	92.9	72.7	88.9
30-day mortality	14 (*n* = 7)	5.5 (*n* = 6)	11.3 (*n* = 8)	7.1 (*n* = 1)	20 (*n* = 3)	2.2 (*n* = 1)
Residual MR						
None/mild	100	99	92.5	93	100	92.7
Moderate/severe	0	1	7.5	7	0	7.3
LVOT obstruction	0	0	0	7.1 (*n* = 1)	6.6 (*n* = 1)	0

Values are mean (range), mean ± SD, median [interquartile range], *n* (%), or *n*. MR: mitral regurgitation; LVEF: left ventricle ejection fraction; LVOT: left ventricle outflow tract; TA: transapical; TF: transfemoral.

## Data Availability

Not applicable.

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
