# Peer review of "Imaging in Transcatheter Mitral Valve Replacement: State-of-Art Review"

_jcm, 2021, doi:10.3390/jcm10245973_

Round 1

Reviewer 1 Report

The present manuscript from Barreiro-Perez et al. is a state-of the-art article for TMVR from imaging perspective, pre- and periporocedural.

The authors should decide if they want to submit the work as original article-review section, or as state-of-the art article. They put it in the section „stat-of-the-art review“, which is not appropriate, as the manuscpript does not fullfil the criteria of original scientific work. In this order, for example, the authors do not provide a methods section – how do they selected the articles, which database have been used, which including criteria. How many published work are available for comparison? There is no scientific question, which should be answered, there is no goal and endpoint of a study.  Even if it is put in the „review“ section, they need to perform the statistical analysis, report on all available data up to date, sort it in a table, and make evidence based conclusions. The maniscpript is more written like a lecture, proffessional opinion, and master statement of state of the art regarding imaging and planing the TMVR procedure. This is a very well written work, which includes systematized and important information. Some paragraphes are redundant and provide double the same information, but all over it could be of great reader’s interest.

I suggest to resubmit the revised manuscript to the „state-of-the-art“/„how-to-do-it“ section of the journal and not to scientifc origninal works section.  

Author Response

We would like to thank Reviewer#1 for this suggestion and we will resubmit our manuscript as “how-to-do-it” section. 

Reviewer 2 Report

General comments.

Perez et al. present a comprehensive review on available imaging techniques for transcatheter mitral valve replacement; the review is well-conducted and clear in its messages giving a good overview of imaging role in transcatheter mitral valve replacement.

English language quality is good. Figures are clear and well-described.

However, some improvements are needed to make the paper publishable.

Specific comments.

Introduction: As treated in the text, valve in native mitral valve replacement should be briefly introduced in this section.

Imaging overview: Line 122; LAMPOON and tip-to base (reverse) LAMPOON should be both mentioned in the text. Moreover, citation needs correction: the one reported refers only to tip-to base (reverse )LAMPOON.

Procedural description: there is lack of citations. Procedural steps are described in detail in several papers that authors should cite (i.e. Guerrero M. et al. ‘Transseptal transcatheter mitral valve-in-valve: A step by step guide from preprocedural planning to postprocedural care’.)

In this section, echo-guided femoral vein puncture should be mentioned: imaging is also helpful in the first step of a transcatheter procedure.

Clinical results and published evidence: Line 179-180; it is a repetition of the sentence in line 138-139.

Imaging key aspects I) Valve-in-Valve: Valve in Valve Mitral app is an useful tool for clinicians in procedural planning that deserves to be mentioned.

Imaging key aspects III) Valve-in-MAC: Authors should underline the difficult patient selection for Valve-in-MAC procedure, as stated in the paper they appropriately cite (Ref.21):  92 patients cases presented in case review call and only 31 patients enrolled; the main reasons for exclusion were risk of left ventricular outflow tract obstruction and risk of embolization: authors should enlighten this point and the precious role of cardiovascular imaging in this specific subset of patients.

In particular, regarding CT evaluation and left ventricular outflow tract obstruction in Valve-in-MAC, findings from recently published in Circulation: Cardiovascular interventions 'Predictors of Left Ventricular Outflow Tract Obstruction After Transcatheter Mitral Valve Replacement in Severe Mitral Annular Calcification: An Analysis of the Transcatheter Mitral Valve Replacement in Mitral Annular Calcification Global Registry' should be reported.

Eventually, a statement that helps readers to understand the difference between valve-in-valve/valve in ring procedures that are well established alternatives to surgery and valve-in-MAC that is still at the ‘beginning of the journey’ and needs lots of improvements to reach a sufficient safety and efficacy profile may complete this section.

VALVE IN NATIVE MITRAL VALVE REPLACEMENT: The recently published and presented at TCT 2021 'Intrepid TMVR Early Feasibility Study Results' should be mentioned in the text and table should be corrected, presenting Intrepid as a valve implantable also via transfemoral-transeptal access.

Author Response

Specific comments.

  • Introduction: As treated in the text, valve in native mitral valve replacement should be briefly introduced in this section.
    • Thank you for your suggestion. We have included a specific reference (line 59)

  • Imaging overview: Line 122; LAMPOON and tip-to base (reverse) LAMPOON should be both mentioned in the text. Moreover, citation needs correction: the one reported refers only to tip-to base (reverse )LAMPOON.
    • We are thankful to Reviwer#2 for this right comment. We have included new references for base-to-tip and anterograde LAMPOON, moreover to correct the citation mistake. “Preemptive LVOT obstruction avoidance strategies have been reported in selected high-risk cases such as alcoholic septal ablation or LAMPOON techniques (base-to-tip (Khan JM, Babaliaros VC, Greenbaum AB, Foerst JR, Yazdani S, McCabe JM et al. Anterior Leaflet Laceration to Prevent Ventricular Outflow Tract Obstruction During Transcatheter Mitral Valve Replacement. J Am Coll Cardiol. 2019;73(20):2521-2534.); tip-to-base or reverse LAMPOON (22), or anterograde LAMPOON (Lisko JC, Greenbaum AB, Khan JM, Kamioka N, Gleason PT, Byku I, et al. Antegrade Intentional Laceration of the Anterior Mitral Leaflet to Prevent Left Ventricular Outflow Tract Obstruction: A Simplified Technique From Bench to Bedside. Circ Cardiovasc Interv. 2020;13(6):e008903. Epub 2020 Jun 9.))”

  • Procedural description: there is a lack of citations. Procedural steps are described in detail in several papers that authors should cite (i.e. Guerrero M. et al. ‘Transseptal transcatheter mitral valve-in-valve: A step by step guide from preprocedural planning to postprocedural care’.)
    • That´s a right comment. We have included in the text a new sentence and two references to step-by-step papers. “Procedural steps are described in detail in published papers (Guerrero M, Salinger M, Pursnani A, Pearson P, Lampert M, Levisay J et al. Transseptal transcatheter mitral valve-in-valve: A step by step guide from preprocedural planning to postprocedural care. Catheter Cardiovasc Interv. 2018;92(3):E185-E196. // Urena M, Himbert D, Brochet E, Carrasco JL, Lung B, Nataf P, Vahanian A. Transseptal Transcatheter Mitral Valve Replacement Using Balloon-Expandable Transcatheter Heart Valves: A Step-by-Step Approach. JACC Cardiovasc Interv. 2017;10(19):1905-1919.)”

  • In this section, echo-guided femoral vein puncture should be mentioned: imaging is also helpful in the first step of a transcatheter procedure.
    • This is a good perception pointed by Reviewer#2, we have added a new paragraph about ultrasound-guided femoral access; “In case of transseptal approach, femoral access is needed. There is a general consensus that ultrasound guidance for obtaining vascular access is considered the standard of care, not only in cardiac interventions but also in electrophysiological procedures (Vincent F, Spillemaeker H, Kyheng M, Belin-Vincent C, Delhaye C, Piérache A et al. Ultrasound Guidance to Reduce Vascular and Bleeding Complications of Percutaneous Transfemoral Transcatheter Aortic Valve Replacement: A Propensity Score-Matched Comparison. J Am Heart Assoc. 2020 Mar 17;9(6):e014916. Epub 2020 Mar 16.)”

  • Clinical results and published evidence: Line 179-180; it is a repetition of the sentence in lines 138-139.
    • This sentence has been erased from the manuscript.

  • Imaging key aspects I) Valve-in-Valve: Valve in Valve Mitral app is a useful tool for clinicians in procedural planning that deserves to be mentioned.
    • It is a really appropriate comment and we have included a mention in the manuscript and both references for GooglePlay and AppleStore. “A smartphone app has been developed, and is available for different platforms, to assist SHV size selection before TMVR ViV (Play.google.com. 2021. Mitral Valve-in-Valve APP. [online] Available at: <https://play.google.com/store/apps/details?id=com.ubqo.vivmitral&hl=es_AR&gl=US> [Accessed 10 December 2021]. //  App Store. 2021. ‎Valve In Valve (Mitral). [online] Available at: <https://apps.apple.com/es/app/valve-in-valve-mitral/id703369667> [Accessed 10 December 2021].)”

  • Imaging key aspects III) Valve-in-MAC:Authors should underline the difficult patient selection for Valve-in-MAC procedure, as stated in the paper they appropriately cite (Ref.21):  92 patients cases presented in case review call and only 31 patients enrolled; the main reasons for exclusion were risk of left ventricular outflow tract obstruction and risk of embolization: authors should enlighten this point and the precious role of cardiovascular imaging in this specific subset of patients.
    • We would like to thank Reviewer#2 for his/her appropriate observation, we added a short sentence emphasizing this aspect. “The appropriate pre-procedural patient selection for Valve-in-MAC requires expertise, is time-consuming and it has to be on consideration several anatomical aspects. In previously published series, only 33% of evaluated cases for ViMAC were finally acceptable for TVMR procedure (21)”.

  • In particular, regarding CT evaluation and left ventricular outflow tract obstruction in Valve-in-MAC, findings from recently published in Circulation: Cardiovascular interventions'Predictors of Left Ventricular Outflow Tract Obstruction After Transcatheter Mitral Valve Replacement in Severe Mitral Annular Calcification: An Analysis of the Transcatheter Mitral Valve Replacement in Mitral Annular Calcification Global Registry' should be reported.
    • We have included this reference in the manuscript and the key findings described on int. Thanks for this suggestion. Besides the neo-LVOT area, other anatomical features have been recently related to neo-LVOT obstruction after TVMR ViMAC; systolic LVOT area, indexed neo-LVOT, expected LVOT area reduction, and virtual THV to septum distance (Sabbagh AE, Al-Hijji M, Wang DD, Eleid M, Urena M, Himbert D, et al. Predictors of Left Ventricular Outflow Tract Obstruction After Transcatheter Mitral Valve Replacement in Severe Mitral Annular Calcification: An Analysis of the Transcatheter Mitral Valve Replacement in Mitral Annular Calcification Global Registry. Circ Cardiovasc Interv. 2021 ;14(10):e010854. Epub 2021 Oct 19.).

  • Eventually, a statement that helps readers to understand the difference between valve-in-valve/valve in ring procedures that are well-established alternatives to surgery and valve-in-MAC that is still at the ‘beginning of the journey and needs lots of improvements to reach a sufficient safety and efficacy profile may complete this section.
    • That’s a nice point for the general reader. Thank you for your comment. “Unlike ViV and ViR, which are more consolidated procedures, included in clinical care and guidelines recommendations; ViMAC in an early phase of development and it should be reserved for selected patients in highly experienced centers”

  • VALVE IN NATIVE MITRAL VALVE REPLACEMENT: The recently published and presented at TCT 2021 'Intrepid TMVR Early Feasibility Study Results' should be mentioned in the text and table should be corrected, presenting Intrepid as a valve implantable also via transfemoral-transeptal access.
    • According to the reviewer, we have added those data in a new paragraph and data in Table 2; “Very recently, 30-day outcomes of an early feasibility trial with a novel TMVR system have been presented (J Am Coll Cardiol Intv Nov 06, 2021. ePublished Thirty-Day Outcomes Following Transfemoral Transseptal Transcatheter Mitral Valve Replacement: Intrepid TMVR Early Feasibility Study Results). The Intrepid TMVR is a novel device designed in order to treat patients with severe MR through femoral access with 35Fr sheath. Initial results despite including a very selected population are promising. In a cohort of advanced-age patients with mainly primary MR and mildly impaired LVEF there are no deaths, strokes or reinterventions at 30-days. Significant improvement in NYHA functional class has been also reported. Nonetheless, around 50% of patients had significant major bleeding events due to access site major vascular complications. There is a promising landscape for this device but at this moment only preliminary data are available.”

Reviewer 3 Report

Dear Authors, thank you for this fine manuscript. However, I would like to encourage you to make some changes which may positively Impact the Quality of this article.

  1. Please discuss more detailed patients characteristics with regard to the mitral Valve who can not undergo surgery. Please also comment on Risk scores evaluating operative Risk.
  2. Please describe more comprehensive the anatomical Features eliglible for TEER.
  3. Please discuss studies comparing surgical mitral Valve repair/replacement with transcatheter mitral Valve repair/replacement
  4. Please comment on re-operation rates after TEER, which theraputical Options can be ruled out after failed TEER?

Author Response

  • Please discuss more detailed patient characteristics with regard to the mitral Valve that cannot undergo surgery. Please also comment on Risk scores evaluating operative Risk.
    • The authors would like to thank Reviewer#3 for these comments. We don´t include an extensive discussion on these terms due to the manuscript is focused on transcatheter mitral valve replacement procedures, a multi-modality imaging overview; and the references included in the manuscript cover these issues.

According to the current guidelines, prior to including any patient for TMV repair or replacement procedures, it should be discussed in a multidisciplinary Heart Team session that includes cardiac surgeons. The guidelines recommended employing scoring systems to assess the surgical risk and to consider as well other relevant comorbidities not included in these scores. There are two established conventional scoring systems (EuroSCORE and STS-Score) for the surgical risk assessment in patients with structural heart diseases. These traditional scoring systems focus on the evaluation of the surgical risk concerning surgery-related mortality and morbidity. Despite the limitations of these scores, both are used in clinical routine, and the STS score was employed for risk assessment in the TEER pivotal trials. New scores as the Mitral Regurgitation International Database (MIDA) Score have been published to treat to predicts short- and long-term outcomes in patients with mitral regurgitation, but it has not been included yet in the guidelines or clinical routine.

As an example of the basal characteristics of patients referred for transcatheter MV replacement procedures, in the TMVR registry all patients were prior refused for surgery, >70 years, 54% females, with a high STS-score 9,0 (+/-7,0), and major comorbidities as prior stroke (16%), CPOD (30%) or prior cardiac surgery (91%; prior CABG 33%)

  • Please describe more comprehensively the anatomical Features eligible for TEER.
    • Attending to the manuscript length, many details of the TEER procedure were not fully covered in the manuscript, due to its main focus being on transcatheter mitral valve replacement. The authors had included a reference where the anatomical features eligible for TEER are detailed. (Beigel R, Wunderlich NC, Kar S et al. The evolution of percutaneous mitral valve repair therapy: Lessons learned and implications for patient selection. J Am Coll Cardiol 2014; 64:2688–2700.).

In concordance with your suggestion, we included as well briefly in the manuscript. Thank you for your comment.

  • Please discuss studies comparing surgical mitral Valve repair/replacement with transcatheter mitral Valve repair/replacement
    • We would like to thank Reviwer#3 for this interesting point for discussion.

The authors focused the manuscript on transcatheter mitral valve replacement, and the results of these techniques are exposed in the text. The evidence available in this field comes from observational studies and registries. The patients recruited for these procedures are previously considered non-operable or at a high-surgical risk, so there is a lack of evidence comparing these TMVR techniques to surgery in similar populations.

Attending to repair techniques, those are out of the field of our review, we could differ between primary and secondary mitral regurgitation. On primary MR, the EVEREST-II trial (DOI: 10.10567nejmOA1009355) is the only trial specifically designed to randomly compare surgical and transcatheter treatments. Although the percutaneous repair was less effective at reducing mitral regurgitation than conventional surgery, the procedure was associated with superior safety and similar improvements in clinical outcomes. It must be mentioned that in the surgical group 14% of patients received a surgical MV replacement instead of a repair, and when this trial was performed (2005-2008) TEER improved devices and a high-operator experience were not available. The current guidelines recommend TEER in primary MR only in non-candidate patients for surgical treatment. 

In the field of secondary MR, a few trials have been published comparing surgical vs. medical, and transcatheter vs. medical treatments; but there is not a direct comparison between surgical and transcatheter options. A recently published propensity score-matched cohorts comparison revealed no difference in two-year survival after transcatheter or surgical MV repair (DOI: 10.1016/j.jtcvs.2021.07.029). However, these results should be taken with caution due to in the matched cohort the surgical risk was not comparable (8,1% transcatheter vs. 5,2% surgical).

  • Please comment on re-operation rates after TEER, which therapeutical Options can be ruled out after failed TEER?
    • The authors are grateful for this nice comment. In the COAPT trial (DOI: 10.1056/NEJMoa1806640), 4% of patients in the device group underwent an unplanned mitral-valve intervention (3.7% re-TEER and 0,4% surgery) after a 24-months follow-up.

Recently, a new option has been reported for a failed TEER in a patient non-candidate for a new TEER procedure or MV surgery. The ELASTA-Clip is a feasible and safe transcatheter electrosurgical detachment of failed TEER clips from the anterior leaflet followed by TMVR. (Lisko JC, Greenbaum AB, Guyton RA, Kamioka N, Grubb KJ, Gleason PT et al. Electrosurgical Detachment of MitraClips From the Anterior Mitral Leaflet Prior to Transcatheter Mitral Valve Implantation. JACC Cardiovasc Interv. 2020;13(20):2361-2370.).

We have included the ELASTA-Clip mention in the manuscript as a novel option for TVMR.

Round 2

Reviewer 1 Report

The authors answered appropriate to the main concern.

Reviewer 3 Report

no further questions